# Predictors of Immunogenic Response to the BNT162b2 mRNA COVID-19 Vaccination in Patients with Autoimmune Inflammatory Rheumatic Diseases Treated with Rituximab

**DOI:** 10.3390/vaccines10060901

**Published:** 2022-06-06

**Authors:** Victoria Furer, Tali Eviatar, Devy Zisman, Hagit Peleg, Yolanda Braun-Moscovici, Alexandra Balbir-Gurman, Daphna Paran, David Levartovsky, Michael Zisapel, Ofir Elalouf, Ilana Kaufman, Adi Broyde, Ari Polachek, Joy Feld, Amir Haddad, Tal Gazitt, Muna Elias, Nizar Higazi, Fadi Kharouf, Sara Pel, Sharon Nevo, Ori Elkayam

**Affiliations:** 1Rheumatology Department, Tel Aviv Sourasky Medical Center, Sackler Faculty of Medicine, Tel Aviv University, Tel Aviv 6423906, Israel; talieviatar@gmail.com (T.E.); parandaphna@gmail.com (D.P.); davidl@tlvmc.gov.il (D.L.); zismichael@hotmail.com (M.Z.); alalufo@gmail.com (O.E.); kaufmanil17@gmail.com (I.K.); adibroyde@yahoo.com (A.B.); arip@tlvmc.gov.il (A.P.); sarap@tlvmc.gov.il (S.P.); sharonne@tlvmc.gov.il (S.N.); oribe14@gmail.com (O.E.); 2Rheumatology Unit, Carmel Medical Center, Haifa 3436212, Israel; devyzisman@gmail.com (D.Z.); joyfeld@gmail.com (J.F.); haddadamir@yahoo.com (A.H.); tgazitt@gmail.com (T.G.); munael@clalit.org.il (M.E.); nizar.m.h@me.com (N.H.); 3Rheumatology Unit, Hadassah Medical Center, Hebrew University of Jerusalem, Jerusalem 9103401, Israel; hagitp@hadassah.org.il (H.P.); fadikharouf@hotmail.com (F.K.); 4Shine Rheumatology Institute, Rambam Health Care Campus, Rappaport Faculty of Medicine, Technion-Israel Institute of Technology, Haifa 3436212, Israel; y_braun@rambam.health.gov.il (Y.B.-M.); a_balbir@rambam.health.gov.il (A.B.-G.)

**Keywords:** mRNA vaccine, COVID-19, SARS-CoV-2, rituximab, immunogenicity

## Abstract

Treatment with rituximab (RTX) blunts SARS-CoV-2 vaccination-induced humoral response. We sought to identify predictors of a positive immunogenic response to the BNT162b2 mRNA vaccine in patients with autoimmune inflammatory rheumatic diseases (AIIRD) treated with RTX (AIIRD-RTX). We analyzed 108 AIIRD-RTX patients and 122 immunocompetent controls vaccinated with BNT162b2 mRNA participating in a multicenter vaccination study. Immunogenicity was defined by positive anti-SARS-CoV-2 S1/S2 IgG. We used a stepwise backward multiple logistic regression to identify predicting factors for a positive immunogenic response to vaccination and develop a predicting calculator, further validated in an independent cohort of AIIRD-RTX BNT162b2 mRNA vaccinated patients (*n* = 48). AIIRD-RTX patients who mounted a seropositive immunogenic response significantly differed from patients who did not by a lower number of RTX courses (median (range) 3 (1–10) vs. 5 (1–15), *p* = 0.007; lower cumulative RTX dose (mean ± SD) 6943.11 ± 5975.74 vs. 9780.95 ± 7240.12 mg, *p* = 0.033; higher IgG level prior to last RTX course (mean ± SD), 1189.78 ± 576.28 vs. 884.33 ± 302.31 mg/dL, *p* = 0.002), and extended interval between RTX treatment and vaccination, 469.82 ± 570.39 vs. 162.08 ± 160.12 days, *p* = 0.0009, respectively. Patients with ANCA-associated vasculitis and inflammatory myositis had a low likelihood of a seropositive immunogenic response compared to patients with rheumatoid arthritis, odds ratio (OR) 0.209, 95% confidence interval (CI) 0.046–0.96, *p* = 0.044 and OR 0.189, 95% CI 0.036–0.987, *p* = 0.048, respectively. Based on these findings, we constructed a calculator predicting the probability of a seropositive immunogenic response following BNT162b2 mRNA vaccination which performed with 90.5% sensitivity, 59.3% specificity, and 63.3% positive and 88.9% negative predictive values. In summary, the predicting calculator could guide clinicians for optimal timing of BNT162b2 mRNA vaccination in AIIRD-RTX patients.

## 1. Introduction

The SARS-CoV-2-provoked COVID-19 pandemic has urged the development and authorization of novel messenger RNA (mRNA) vaccines that were proven to be immunogenic and effective among the immunocompetent population [1,2]. Patients with autoimmune inflammatory rheumatic diseases (AIIRD) have been prioritized to receive COVID-19 vaccination [3]. Since immunosuppressed patients were excluded from the landmark vaccine trials, there was uncertainty in the medical community about the response of AIIRD patients to vaccination. Several prospective controlled studies, however, reassuringly proved that most patients with rheumatic diseases could mount an adequate immunogenic response to SARS-CoV-2 vaccination, acknowledging a lower post-vaccination level of anti-SARS-CoV-2 antibodies in AIIRD patients compared to immunocompetent controls [4,5,6,7,8,9].

Among other factors, B-cell-depleting therapy significantly contributes to a reduced immunogenic response to SARS-CoV-2 vaccination in patients with AIIRD [10,11]. Indeed, B-cell depleting therapy, such as rituximab, has been consistently associated with reduced immunogenicity induced by influenza and pneumococcal vaccines [12]. Rituximab (RTX), a widely used B-cell-depleting therapy for rheumatoid arthritis (RA), anti-neutrophil cytoplasmic antibody (ANCA)-associated vasculitis (AAV), and other rheumatic diseases, was identified as a risk factor for COVID-19 complications and a severe disease course [10,13,14,15], emphasizing the importance of SARS-CoV-2 vaccination in vulnerable RTX-treated patients. On the other hand, RTX was linked to a reduced immunogenic response to SARS-CoV-2 vaccination, corresponding to a low seroconversion rate ranging between 24% and 49% [6,7,16,17,18,19]. Furthermore, B-cell-depleting therapy was reported to impair not only humoral but also cell-mediated immune response to SARS-CoV-2 mRNA vaccination [20], although other studies demonstrated a preserved T-cell-mediated immune response in the majority of RTX-treated patients, irrespective of the humoral response [8,17,21]. However, the extent of anti-COVID-19 protection conferred by T-cell-mediated immune response in RTX-treated AIIRD (AIIRD-RTX) patients remains unknown.

In the setting of the COVID-19 pandemic, clinicians commonly face a challenge concerning the optimal timing of vaccination in relation to RTX treatment, despite a general recommendation to delay the B-cell-depleting therapy in relation to SARS-CoV-2 vaccination [22]. Identifying routine available predictors of a seropositive immunogenic response to SARS-CoV-2 vaccination in AIIRD-RTX patients may assist in a patient-tailored vaccination approach. In support of this approach, several studies found a direct correlation between detectable CD19 peripheral B cell counts and an immunogenic response to mRNA SARS-CoV-2 vaccination [17,20]. However, routine measurement of CD19 B cells prior to RTX treatment has not been recommended for most AIIRD and, therefore, might be unavailable in daily practice. Our prospective study on a large cohort of AIIRD patients and immunocompetent controls revealed that the time interval between RTX administration and vaccination had a critical role in predicting an immunogenic response to the BNT162b2 mRNA vaccine [6]. Therefore, we now sought to investigate additional predictors associated with an immunogenic response to mRNA BNT162b2 vaccination in the AIIRD-RTX patients participating in our ongoing prospective vaccination study. We further developed a calculator based on clinical and laboratory data available in daily clinical practice to predict a seropositive immunogenic response conferred by SARS-CoV-2 vaccination in AIIRD-RTX patients.

## 2. Materials and Methods

This study is part of an ongoing prospective observational multicenter study investigating immunogenicity and safety of the BNT162b2 mRNA COVID-19 vaccine in adult AIIRD patients, focusing upon RTX-treated patients compared to immunocompetent controls. The study was conducted at the Rheumatology Departments of the Tel Aviv Sourasky, Carmel, and Hadassah Medical Centers, Israel, between December 2020 and June 2021. The study protocol was described in detail elsewhere [6]. Data from an independent cohort of RTX-treated BNT162b2 mRNA vaccinated AIIRD patients from the Rambam Medical Health Care Campus were used for validation of the prediction model.

The study was performed in accordance with the principles of the Declaration of Helsinki and approved by the research ethics committees of the four participating medical centers: TLV-1055-20, CMC-0238-20, HMO-0025-21, and RMB-417-20, respectively. All study participants gave written informed consent upon recruitment into the study.

### 2.1. Study Aims

The primary endpoint was to identify independent predictors associated with seropositive immunogenic response to the BNT162b2 (Pfizer-BioNTech) vaccine in adult AIIRD-RTX patients.

### 2.2. Secondary Endpoints

Immunogenicity of the BNT162b2 mRNA vaccine in adult AIIRD-RTX patients compared with immunocompetent controls.Development of a calculator to predict the probability of a seropositive immunogenic response to the BNT162b2 mRNA vaccination in AIIRD-RTX patients and its validation in an independent cohort of vaccinated AIIRD-RTX patients.Safety of the vaccination.Effect of the vaccination on disease activity in AIIRD-RTX patients stratified by positive and negative immunogenic response to vaccination.

### 2.3. Study Population

This study included consecutive 86 AIIRD patients treated with RTX up to 8 years prior to the BNT162b2 vaccination, who participated in the vaccination study reported earlier by our group [6], and 22 additional suitable RTX-treated patients (Figure 1). All consenting adult (≥18 years of age) AIIRD patients who fulfilled the ACR/EULAR criteria for RA [23], Systemic Lupus International Collaborating Clinics classification criteria for systemic lupus erythematosus (SLE) [24], Chapel Hill classification criteria for systemic vasculitis (AAV, giant cell arteritis, other systemic vasculitides) [25], and the EULAR/ACR classification criteria for idiopathic inflammatory myopathy (IIM) [26] were recruited. The control group was composed of the immunocompetent individuals, participating as controls in the earlier study (*n* = 122). An independent prospectively recruited cohort of BNT162b2 mRNA vaccinated AIIRD-RTX patients (*n* = 48) from the Rambam Medical Health Care Campus was used for the validation of the prediction model.

Exclusion criteria for all study participants were pregnancy, a history of past vaccination allergy, and a previous COVID-19 infection, and those for the controls were a history of AIIRD, history of immunosuppressive treatment, and a previous COVID-19 infection.

### 2.4. Data Collection

Demographic and clinical characteristics, including AIIRD diagnosis and anti-rheumatic medications, were reported by the participants and confirmed by reviewing the electronic medical records (EMR) by the study investigators (VF, TE, DZ, HP). The dates of the seasonal 2020 influenza vaccination and the BNT162b2 mRNA vaccination were recorded. The participants’ medications included conventional synthetic disease modifying antirheumatic drugs (csDMARDs), glucocorticoids (GC), other immunosuppressive medications (e.g., mycophenolate mofetil), and intravenous immunoglobulin (IVIg). Doses of methotrexate (MTX) and prednisone were recorded. Specific details regarding The RTX treatment data that were collected from the EMR included immunoglobulin G levels (IgG, mg/dL) up to 3 months prior to the last RTX course, the total number of RTX courses regardless of indication, the dose of each RTX course and the cumulative RTX dose (mg), and the date of the last RTX course. Hypogammaglobulinemia was defined as a total IgG level (i.e., prior to the last RTX course) of less than 500 mg/dL. The time interval between the last RTX course and the BNT162b2 vaccination was calculated in days. Data on the CD19-positive B cell counts at the time of RTX administration prior to the vaccination were unavailable.

### 2.5. Vaccination Procedure

All participants were administered the two-dose regimen of the BNT162b2 Pfizer BioNTech mRNA vaccine according to national guidelines. Each 30 µg dose was given as an intramuscular injection in the deltoid muscle, and the second dose was given 3 weeks after the first one.

### 2.6. Vaccine Immunogenicity

All study participants underwent a serological test from 2 to 6 weeks after the second vaccine dose. SARS-CoV-2 S1/S2 IgG antibodies were measured by the Food and Drug Administration (FDA)-authorized LIAISON (Diasorin, Saluggia (VC), Italy) quantitative assay, with 98% sensitivity and specificity [27]. A cut-off of 15 binding antibody units (BAU) was considered a positive immunogenic response according to the manufacturer’s instructions. In the validation group, anti-spike receptor-binding domain (RBD) antibodies were measured by the SARS-CoV-2 IgG II Quant (Abbott) assay (a chemiluminescent microparticle immunoassay) on the ARCHITECT ci8200system (Abbott, IL, USA). This test was considered positive when titers were above 50 AU/mL, according to the manufacturer’s instructions. An inter-assay validation analysis between the two assays was beyond the scope of this study, however, published data suggest good diagnostic performance and strong correlations with neutralizing antibodies for both [28,29].

### 2.7. Vaccine Safety

The study participants were queried (by telephone or in person) about adverse events 2 weeks after the 1st vaccine dose and 2–6 weeks after the 2nd vaccine dose. Adverse events were reported when they occurred at the injection site or when they occurred in temporal proximity to vaccination.

### 2.8. Assessment of AIIRD Activity

Information on pre-vaccination disease activity during the 3 months preceding vaccination was retrieved from the medical records. Post-vaccination disease activity was clinically assessed 2–6 weeks after the 2nd dose. The disease activity indices that were used for RA included the Clinical Disease Activity Index (CDAI), Simplified DAI (SDAI), and DAS-28-CRP, while the Systemic Lupus Disease Activity Index (SLEDAI) was used for SLE, and the Patient and Physician Global Assessment (PGA and PhGA, respectively, by means of a visual analogue scale [VAS] of 0–10 mm) was used for vasculitis and IIM.

### 2.9. Statistical Analysis

Differences between categorical variables were tested with Fisher’s exact test. Differences between numeric variables were tested with the *t*-test. A stepwise backward multiple logistic regression for predicting a seropositive response to vaccination was applied to AIIRD patients for whom all data were available (*n* = 104). The AIIRD diagnosis was a dummy variable, meaning that each participant could have only one diagnosis. The model included all individual variables that showed a significance level of *p* < 0.2 between seropositive and seronegative results. The rule for leaving the variable in the model was *p* < 0.2. Multicollinearity between significant variables was assessed by Pearson correlations. The prediction calculator was tested on an independent cohort of 48 BNT162b2 vaccinated AIIRD-RTX patients (the validation group), including 21 patients with a positive immunogenic response (responders) to vaccination and 27 patients without a detectable immunogenic response (non-responders) to vaccination. The calculated fixed values for the tested population were plotted on a receiver operator characteristic (ROC) curve to select the optimal discriminative cut-off to predict a positive response to vaccination. Sensitivity, specificity, positive predictive, and negative predictive values (PPV and NPV, respectively) were calculated based on this optimal cut-off of the ROC curve.

## 3. Results

### 3.1. Study Participants

The study included 108 AIIRD-RTX patients and 122 controls, all vaccinated with the 2-dose regimen of the BNT162b2 mRNA vaccine. Demographics and clinical data of the AIIRD-RTX population are presented in Table 1. The AIIRD-RTX patients were significantly older than the controls (mean ± standard deviation [SD] 61.45 ± 14.96 vs. 50.83 ± 14.64 years, *p* < 0.0001). The majority were females in both the patient and control groups (76.85% [*n* = 83] and 64.75% [*n* = 79], *p* = 0.06), with a high uptake of the seasonal 2020 influenza vaccination prior to the BNT162b2 mRNA vaccination (84.11% [*n* = 90] and 81.65% [*n* = 89], *p* = 0.72, respectively). The most common AIIRD diagnosis was RA (45.37%, *n* = 49), followed by AAV (21.3%, *n* = 23), IIM (16.67%, *n* = 18), SLE (10.19%, *n* = 11), and other vasculitides (5.66%, *n* = 6).

### 3.2. Immunogenicity of the BNT162b2 Vaccine

The BNT162b2 vaccine-induced positive immunogenic response rate and serum S1/S2 IgG titers were significantly lower in the AIIRD-RTX group compared to controls (41.67% [*n* = 45] vs. 100% [*n* = 122], *p* < 0.0001 and 51.01 ± 79.17 vs. 218.39 ± 81.76 BAU, *p* < 0.0001, respectively). The lowest S1/S2 IgG titer (BAU, mean ± SD) was detected in the AAV and IIM patients (36.25 ± 73 and 25.19 ± 45.07, respectively), followed by patients with other non-AAV systemic vasculitides (48.8 ± 74.29), whereas the highest titers were detected in patients with SLE and RA (99.84 ± 110.55 and 55.19 ± 81.55, respectively) (Figure 2). Both vaccine responders and non-responders were similar in age, sex, and concomitant immunosuppressive medication use. The rate of seropositive and seronegative vaccine response was similarly distributed across all rheumatic diseases, except for the SLE patients who had a high prevalence of a seropositive immunogenic response (81.82% [*n* = 9] vs. 18.18% [*n* = 2], *p* = 0.007, respectively). Collectively, the SLE patients had the lowest cumulative RTX dose, and the highest mean IgG levels prior to the last RTX course, as well as a longer interval between the last RTX course and the BNT162b2 vaccine (Table 2 and Appendix A).

Several predictors were associated with a seropositive immunogenic response to the BNT162b2 vaccine among the AIIRD patients treated with RTX. Vaccine responders significantly differed from non-responders by the following parameters (Table 2): (1) higher total IgG level prior to last RTX course (1189.78 ± 576.28 vs. 884.33 ± 302.31 mg/dL, *p* = 0.002, respectively); (2) lower cumulative RTX dose (6943.11 ± 5975.74 vs. 9780.95 ± 7240.12 mg, *p* = 0.033, respectively); (3) lower total number of RTX courses (median [range] 3 [1–10] vs. 5 [1–15], respectively, *p* = 0.007). The time interval between RTX treatment and vaccination was more than two times longer in responders (469.82 ± 570.39 vs. 162.08 ± 160.12 days, *p* = 0.0009, respectively).

In the stepwise backward logistic regression model predicting seropositive immunogenic response to vaccination (Table 3), AAV and IIM diagnoses significantly decreased the likelihood of the response (odds ratio (OR) 0.209, 95% confidence interval (CI) 0.046–0.96, *p* = 0.044, and OR 0.189 95% CI 0.036–0.987, *p* = 0.048, respectively), while higher serum total IgG levels prior to the last RTX course and a longer time interval between RTX treatment and BNT162b2 vaccine increased the likelihood of a response (OR 1.1 95% CI 1.019–1.196 for each IgG level increment by 50 mg/dL, *p* = 0.016, and OR 1.048 95%CI 1.018–1.079 for each passing week after last RTX course, *p* = 0.002, respectively) (Figure 3). There was a moderate correlation between the IgG levels and the total number of RTX courses, but not strong enough to assume multicollinearity, thereby permitting the consideration of both variables as independent predictors.

### 3.3. Development and Validation of the Predicting Calculator for a Seropositive Immunogenic Response to the BNT162b2 mRNA Vaccine in RTX-Treated Patients

We used stepwise backward multiple logistic regression to identify the following independent predictors for a seropositive immunogenic vaccine response: AIIRD diagnosis, total number of RTX courses prior to vaccination, serum IgG levels prior to the last RTX course administered before vaccination, and time interval between the last RTX course and the date of vaccination. Notably, the MTX and CS variables were not significant according to the model. We further developed a prediction calculator using the equation presented in Figure 4.
$$\frac{exp\left(-2.682+1.441\cdot SLE-1.665\cdot IIM-1.564\cdot AAV-0.738\cdot Vasculitis+0.002\cdot IgG-0.135\cdot total\;courses+0.007\cdot Days\right)}{1+exp\left(-2.682+1.441\cdot SLE-1.665\cdot IIM-1.564\cdot AAV-0.738\cdot Vasculitis+0.002\cdot IgG-0.135\cdot total\;courses+0.007\cdot Days\right)}$$

Legend: *AAV*, ANCA-associated vasculitis; *IIM*, idiopathic inflammatory myositis; *IgG*, immunoglobulin G; *SLE*, systemic lupus erythematosus. Vasculitis includes non-AAV types of vasculitis.

Next, we validated the model using data from an independent AIIRD-RTX cohort (*n* = 48) vaccinated with the BNT162b2 mRNA vaccine, including 21 responders and 27 non-responders. The characteristics of the validation and main study groups were similar for age, sex, and the rate of seropositive response to vaccination (Appendix A). In contrast to the main study group, there were only three cases of AAV in the validation group. We further applied a ROC curve to select the optimal discriminative cut-off value of 0.41, with a sensitivity of 90.5%, specificity of 59.3%, PPV 63.4%, and NPV 89.9.1% (Figure 4). An example of the calculator use would be an AAV patient with a treatment history of 4 RTX courses, an IgG level of 600 mg/dL, and a 100-day interval until a planned vaccination would have a very low probability (5.3%) of a seropositive response; thus, suggesting the need to test a serologic response following vaccination and consider a vaccination booster later on. In contrast, an RA patient with a treatment history of 2 RTX courses, an IgG level of 1100 mg/dL, and a 100-day interval to a planned vaccination would have a greater likelihood for a seropositive response (estimated as being 42%).

### 3.4. BNT162b2 Vaccine Safety in Seropositive and Seronegative RTX-AIIRD Patients

Vaccine responders and non-responders had a similar profile and rate of vaccine-related adverse events (Appendix A). One of the non-responders died due to fulminant hemorrhagic cutaneous vasculitis with subsequent fatal sepsis. She had AAV in clinical remission following RTX treatment in October 2017 and was being treated solely with low-dose prednisone (5 mg/day) at the time of vaccination [6].

### 3.5. BNT162b2 Vaccine Impact on Disease Activity in RTX-AIIRD Patients

Following vaccination, the disease activity indices of RA patients worsened in 26.5–32.6% of the patients, they were stable in 52.9–60%, and improved in 12.5–20.6%, depending upon the score used (Appendix A) The pattern of disease activity changes was similar for the RA patients who mounted a seropositive immunogenic response to vaccination and those who did not. SLE disease activity as measured by SLEDAI remained stable for 8 of the 9 patients whose pre- and post-vaccination SLEDAI scores were available. The PGA and PhGA-VAS scores before and after vaccination were generally stable in the AAV and IIM patients as well as in those with other vasculitides.

## 4. Discussion

The optimal timing of COVID-19 vaccination in AIIRD patients treated with RTX remains debatable. Herein, we report the findings of an analysis of a large group of RTX-treated AIIRD patients (*n* = 108) vaccinated with the 2-dose BNT162b2 mRNA vaccine regimen who represent a subset of the ongoing multicenter controlled vaccination trial being conducted within a nationwide vaccination campaign.

An immunogenic serologic response against the BNT162b2 mRNA vaccine was observed in all immunocompetent controls, but it was present in only 41.7% of the RTX-treated patients, who had significantly lower post-vaccination anti-spike S1/S2 IgG antibody levels. This finding is in line with previous studies, further confirming the negative impact of RTX on vaccine-induced immunogenicity [6,7,8,9,17,18,19,20,30,31,32]. The safety profile of vaccination was similar among patients with positive and negative immunogenic responses to vaccination and consistent with the findings of our earlier study [6].

We further identified predictors for a positive immunogenic response to vaccination that had been defined in a binary mode by the positive versus negative humoral response measured by anti-spike S1/S2 antibodies. Those predictors included the diagnosis of RA as differentiated from AAV and IIM, a low number of total RTX courses prior to vaccination, high serum total IgG levels prior to the last RTX course, and a longer interval between RTX treatment and vaccination. Recent studies reached the consensus that CD19 reconstitution at the time of vaccination plays a critical role in mounting the immunogenic response to vaccination in RTX-treated patients [8,17,20,31,32,33]. Mrak et al. reported that among 74 RTX-treated AIIRD patients immunized with mRNA vaccines, only those with measurable peripheral B cells developed an immunogenic response to vaccination as measured by antibodies to the receptor-binding-domain (RBD) of the spike protein and by the number of peripheral B cells that correlated with the levels of anti-RBD antibodies [17]. Similar results were reported by Prendecki et al. based on 44 RTX-treated AIIRD patients [8]. Since data on CD19 peripheral B cells were not available in our study and are commonly unavailable in routine practice, we chose to use commonly collected variables to develop the predicting calculator, a transformation of the logistic regression model. It is plausible to assume that the extent of exposure to RTX, as reflected by the total number of RTX courses and the time interval between RTX treatment and vaccination, indicate the extent of B cell reconstitution at the time of vaccination. Indeed, several studies also confirmed the impact of the time interval between RTX treatment and vaccination on predicting the serological response to vaccination [20,31].

Remarkably, AAV and IIM conferred the risk for a negative or poor immunogenic response to vaccination as opposed to RA and SLE, despite the highest cumulative dose of RTX being observed in patients with RA. A potential explanation may relate to the long-lasting RTX-induced depletion of B cells reported in patients with AAV in contrast to patients with RA and connective tissue diseases [34]. After a single course of RTX for the induction of remission, AAV patients might have a very long-lasting B cell depletion of up to more than 60 months, especially patients with microscopic polyangiitis and advanced renal failure [35], indicating a profound dysregulation of the B-cell compartment in these patients. In addition, patients with AAV and IIM are commonly treated with several immunosuppressants for an extended period of time; thus, contributing to an impaired immunogenic response to vaccination. The history of the previous use of immunosuppressants, such as cyclophosphamide or mycophenolate mofetil, was not accounted for in the present model.

The high clinical relevance of our study derives from the development of an algorithm available as a simple to use calculator for predicting the probability of a seropositive immunogenic response to vaccination in RTX-treated AIIRD patients, validated in a separate independent cohort of RTX-treated AIIRD patients (*n* = 48). The calculator is suitable for AIIRD patients with RA, AAV, and other systemic vasculitides, as well as SLE and IIM. RTX-related details required for the calculator use include the total number of RTX courses, the serum IgG level prior to the last RTX course, and the interval between the last RTX course and vaccination. Therefore, the calculator can be helpful in daily practice in the absence of a peripheral B cell count. As the model has a high sensitivity and a good negative predictive value, it is particularly useful for predicting the failure to respond to vaccination; thus, suggesting to postpone vaccination. In view of a relatively low positive predictive value, the prediction of the seropositive immunogenic response might be inaccurate.

Our study has several limitations. First, it should be emphasized that the seropositive immunogenic response by itself—defined as the main outcome of this study—does not necessarily imply protection from COVID-19 infection, especially in the case of low antibody titers [36]. Since there is no consensus on the protective antibody levels in vaccinated patients with rheumatic diseases, however, the use of a seropositive versus a seronegative immunogenic response may be used in clinical practice. Second, there are a number of limitations pertaining to the model of the predicting calculator. We observed a difference in immunogenic response between RA and AAV/IIM patients treated by RTX that had not been reported in other studies, possibly due to differences in cumulative immunosuppression not captured in our model. The moderate diagnostic power of the model can be explained, at least partially, by the small size of the validation cohort that was characterized by a different representation of AIIRD compared to the main cohort. The fact that the immunogenic response was evaluated by different laboratory assays in both cohorts should not significantly affect the results since both assays correlated well with anti-SARS-CoV-2 neutralizing antibody levels [27,28]. Another limitation is that no data on CD19 peripheral cells were available, but the clinical variables we used in the predicting model indirectly reflect that parameter and they are more accessible in daily practice. While the assessment of a T-cell-induced immune response to vaccination may be important in patients with an absent immunogenic response, such an analysis was beyond the scope of this study. Finally, this study was performed with one vaccine type and the findings may not be applicable for other vaccines.

## 5. Conclusions

This large multicenter study based on a diverse AIIRD population treated with RTX identified clinical predictors for a seropositive immunogenic vaccination response to BNT162b2 vaccination: a diagnosis of RA (as opposed to AAV and IIM), a low number of total RTX courses prior to vaccination, a high serum total IgG level prior to the last RTX course, and an extended interval between the RTX treatment and vaccination. Based on these variables, a prediction calculator for assessing the probability of seropositive response to the BNT162b2 vaccination was developed and validated, with a high sensitivity and good negative predictive value. The calculator can be useful for predicting the failure to respond to vaccination; thus, suggesting to postpone vaccination. Taken together, the use of the predicting calculator might guide clinicians for optimal timing of BNT162b2 mRNA vaccination in AIIRD-RTX patients. Further investigation of predictors of vaccine-induced response and vaccine efficacy as well as validation of the proposed prediction algorithm in prospective cohorts are warranted.

## Figures and Tables

**Figure 1 vaccines-10-00901-f001:**
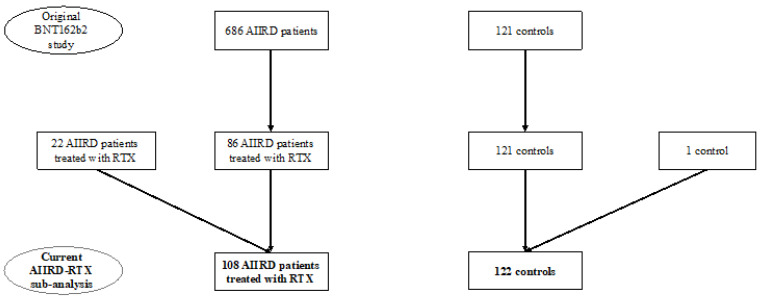
Study flow-chart.

**Figure 2 vaccines-10-00901-f002:**
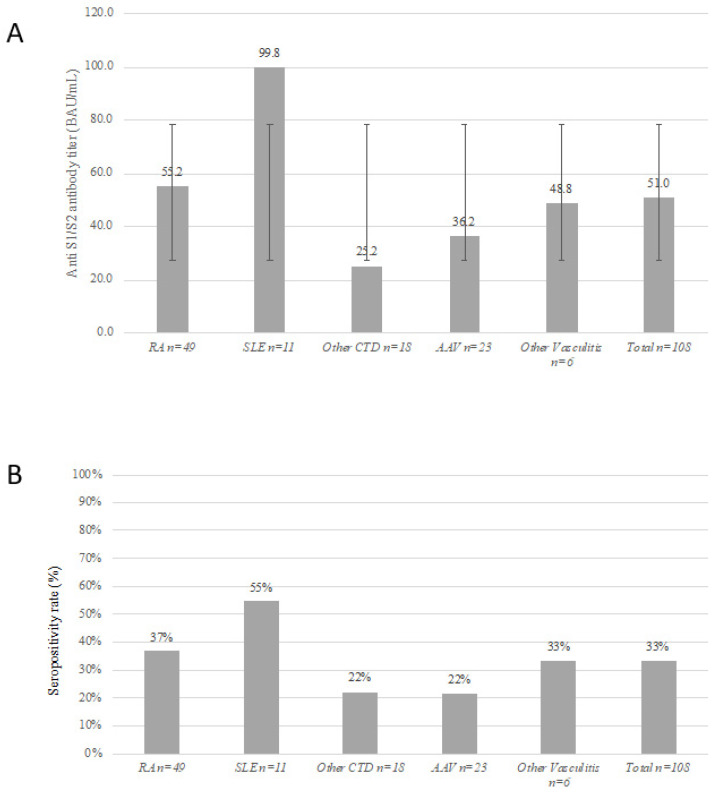
Immunogenic response to the two-dose regimen of BNT162b2 vaccine in patients treated with rituximab according to AIIRD diagnosis. (Panel **A**) Anti S1/S2 IgG titers (mean ± standard deviation). (Panel **B**) Rate of a seropositive response (%). Legend: AIIRD, autoimmune inflammatory rheumatic disease; Ig, immunoglobulin; RA, rheumatoid arthritis; SLE, systemic lupus erythematosus; CTD, connective tissue disease; AAV, ANCA-associated vasculitis.

**Figure 3 vaccines-10-00901-f003:**
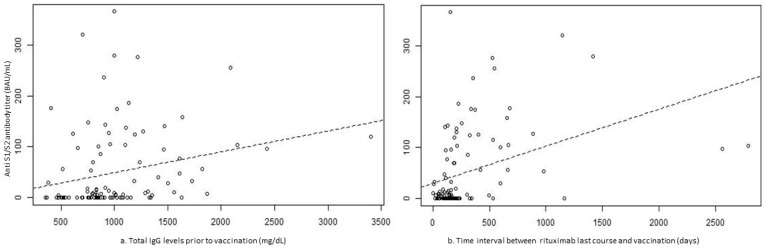
Scatterplots showing a correlation between the immunoglobulin G levels (mg/dL) prior to last rituximab course (panel **a**) and the time interval (days) between the last rituximab course and vaccination (panel **b**) and anti-S1/S2 antibody titer after BNT162b2 mRNA vaccination. Legend: Ab, antibody; BAU, binding antibody units; IgG, immunoglobulin G.

**Figure 4 vaccines-10-00901-f004:**
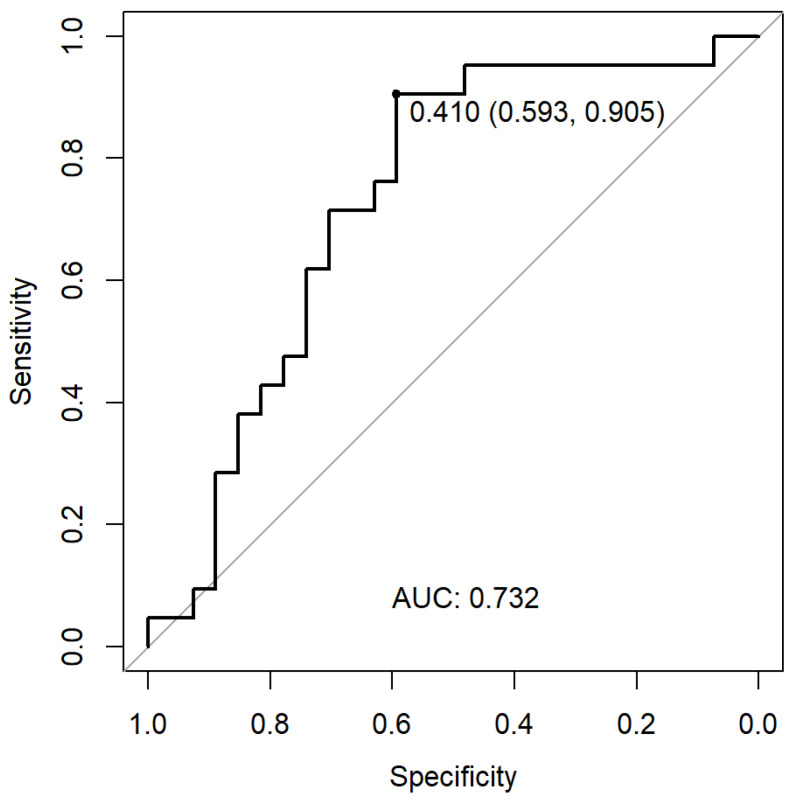
ROC curve for determining the model’s optimal cut-off for a seropositive immunogenic response after BNT162b2 mRNA vaccination in rituximab-treated AIIRD patients.

**Table 1 vaccines-10-00901-t001:** Demographic and clinical characteristics of AIIRD patients treated with rituximab vaccinated with the BNT162b2 mRNA COVID-19 vaccine (*n* = 108).

Age, Years, Median (Range)	65.5 (23–88)
Female sex, *n* (%)	83 (76.85)
Seasonal influenza vaccine uptake, *n*/total (%)	90/107 (84.11)
AIIRD diagnosis, *n* (%)	
RA	49 (45.37)
SLE	11 (10.19)
ANCA-associated vasculitis	23 (21.3)
Other systemic vasculitis	6 (5.56)
IIM	18 (16.67)
Concomitant lymphoma, *n*/total (%)	5/88 (5.68)
AIIRD duration, years, median (range)	10.5 (0.75–45)
Rituximab-relevant details, (mg), median (range)	
Serum IgG level prior last RTX course (mg/dL) (*n* = 105)	911 (357–3405)
Hypogammaglobulinemia < 500 mg/dL (prior to last RTX course), *n*/total (%)	6/106 (5.66)
RTX cumulative dose	6000 (1000–30,000)
RTX dose of last course prior to vaccination	2000 (500–3210)
Total number of RTX courses	4 (1–15)
Interval between last RTX course and BNT162b2 vaccination, days	162.5 (2–2794)
Concomitant immunosuppressive medications, *n* (%)	
csDMARDs	34 (31.48)
Methotrexate	16 (14.81)
Methotrexate dose, mg/week, mean ± SD	13.1 ± 5.32
Prednisone	54 (50)
Prednisone dose, mg/d, mean ± SD	5.9 ± 3.44
Other immunosuppressants, *n* (%)	
Leflunomide	2 (1.85)
Mycophenolate mofetil	5 (4.63)
IVIG	9 (8.33)

Legend: AIIRD, autoimmune inflammatory rheumatic diseases; ANCA, antineutrophil cytoplasmic antibody; csDMARDs, conventional synthetic disease modifying antirheumatic drugs; IIM, idiopathic inflammatory myopathy; IgG, immune globulin G; IVIG, intravenous immune globulin; n, number; RA, rheumatoid arthritis; RTX, rituximab; SD, standard deviation; SLE, systemic lupus erythematosus.

**Table 2 vaccines-10-00901-t002:** Comparison of AIIRD patients treated with rituximab who did or did not mount a positive immunogenic response following vaccination with the BNT162b2 mRNA COVID-19 vaccine.

	Vaccine Responders, *n* = 45	Vaccine Non-Responders, *n* = 63	*p* Value
Age, years, median (range)	64 (29–88)	67 (23–87)	0.075
Sex, female, *n* (%)	37 (82.22)	46 (73.02)	0.356
Seasonal influenza vaccine uptake, *n*/total (%)	35/44 (79.55)	55/63 (87.3)	0.296
Disease duration, years, median (range)	13 (0.75–45)	9 (1–42)	0.146
AIIRD diagnosis, *n* (%) *			
RA	23 (51.11)	26 (41.27)	0.333
SLE	9 (20)	2 (3.17)	0.007
ANCA-associated vasculitis	6 (13.33)	17 (26.98)	0.101
Other systemic vasculitis	2 (4.44)	4 (6.35)	1
IIM, *n* (%)	4 (8.89)	14 (22.22)	0.074
History of lymphoma, *n*/total (%)	4/39 (10.26)	1/49 (2.04)	0.166
Rituximab-relevant details, (mg), median (range)			
Serum IgG level prior last RTX (mg/dL) **	1189.78 ± 576.28	884.33 ± 302.31	0.002
Hypogammaglobulinemia < 500 mg/dL (prior to last RTX course), *n*/total (%)	1/44 (2.27)	5/62 (8.06)	0.397
RTX cumulative dose	4000 (2000–20,000)	8000 (1000–30,000)	0.033
RTX dose of last course prior to vaccination	2000 (500–3210)	2000 (500–2000)	0.168
Total number of RTX courses	3 (1–10)	5 (1–15)	0.007
Time interval between last RTX course and BNT162b2 vaccination, days	255 (6–2794)	130 (2–1163)	0.0009
Up to 180 days, *n* (%)	14 (31.11)	49 (77.78)	<0.0001
181–365 days, *n* (%)	13 (28.89)	11 (17.46)	
Over 365 days, *n* (%)	18 (40)	3 (4.76)	
Concomitant immunosuppressive medication, *n* (%)			
csDMARDs	18 (40)	16 (25.4)	0.142
Methotrexate	6 (13.33)	10 (15.87)	0.789
Methotrexate dose, mg/week, mean ± SD	10.63 ± 4.27	14.17 ± 5.59	0.287
Prednisone	18 (40)	36 (57.14)	0.118
Prednisone dose, mg/d, mean ± SD ***	5.47 ± 3.23	6.16 ± 3.56	0.505
Other immunosuppressants, *n* (%)			
Leflunomide	1 (2.22)	1 (1.59)	1
Mycophenolate mofetil	1 (2.22)	4 (6.35)	0.399
IVIG	3 (6.67)	6 (9.52)	0.732

* Fisher exact test *p*-value for the comparison between the relevant diagnosis (RA, SLE, etc.) and all other diagnoses. ** Data were missing for 3 patients. *** Data were missing for 1 patient. Legend: AIIRD, autoimmune inflammatory rheumatic diseases; ANCA, antineutrophil cytoplasmic antibody; csDMARDS, conventional synthetic disease modifying antirheumatic drugs; IIM, idiopathic inflammatory myopathy; IgG, immune globulin G; IVIG, intravenous immune globulin; n, number; RA, rheumatoid arthritis; RTX, rituximab; SD, standard deviation; SLE, systemic lupus erythematosus.

**Table 3 vaccines-10-00901-t003:** Stepwise backward logistic regression predicting seropositive immunogenic response following BNT162b2 mRNA vaccination (*n* = 104).

Predictors	OR	95% CI	*p* Value
RA	Ref	Ref	Ref
AIIRD diagnosis			
SLE	4.225	0.543–32.89	0.169
ANCA-associated vasculitis	0.209	0.046–0.96	0.044
Other systemic vasculitis	0.478	0.044–5.244	0.546
IIM	0.189	0.036–0.987	0.048
Rituximab-relevant details			
Serum IgG level (50 mg/dL increments, prior to last RTX course)	1.104	1.019–1.196	0.016
Total number of RTX courses	0.874	0.75–1.018	0.084
Time interval between last RTX course and BNT162b2 vaccine (weeks)	1.048	1.018–1.079	0.002

## Data Availability

Not applicable.

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
