# Peer review of "Predictors of Immunogenic Response to the BNT162b2 mRNA COVID-19 Vaccination in Patients with Autoimmune Inflammatory Rheumatic Diseases Treated with Rituximab"

_vaccines, 2022, doi:10.3390/vaccines10060901_

Round 1
Reviewer 1 Report
In this article, the authors analyzed 108 patients with autoimmune inflammatory rheumatic diseases (AIRD) treated with rituximab and 122 immunocompetent controls vaccinated with BNT162b2, and identified a predictor for vaccine immunogenicity to COVID vaccination. The topic is interesting. Some comments are as follows:
- Methods should be better specified. How was the cohort constituted? Were there any drop-out or missing data? Maybe a Flow-chart of the selection process would be helpful.
- The Authors should specify how they considered an AE to be related to vaccination.
- Given age or gender as an independent factor associated with immunogenicity to COVID-19 vaccination, it would be better if the authors are reporting biological sex or gender in your research.
- It would be interesting that the authors could perform survival analysis such as Kaplan-Meier analysis and Cox regression analysis based on an optimal cut-off value of predictor as the proposed prediction algorithm in this research.
- As described in “Methods” section, an independent prospective cohort (n=48) from the Rambam Medical Health Care Campus was used for the validation of the proposed prediction model. The authors may present the related results and have a relevant discussion.
- It would be clearly that the authors could add an additional figure to illustrate the comparisons of anti-S/RBD-IgG positive rates and titers among different groups.
- Some typing errors were noted. For example, in page 5, line 221-222, “Predictors associated with seropositive immunogenic response to the BNT162b2 vaccine in AIRD patients treated with RTX”.
Author Response
In this article, the authors analyzed 108 patients with autoimmune inflammatory rheumatic diseases (AIRD) treated with rituximab and 122 immunocompetent controls vaccinated with BNT162b2, and identified a predictor for vaccine immunogenicity to COVID vaccination. The topic is interesting. Some comments are as follows:
1. Methods should be better specified. How was the cohort constituted? Were there any drop-out or missing data? Maybe a Flow-chart of the selection process would be helpful.
Authors’ reply:
This study represents a sub-analysis of the main vaccination study published by our group (ref: Furer V Ann. Rheum. Dis. 2021, 80, 1330–1338, doi:10.1136/annrheumdis-2021-220647.) The majority of patients and controls participated in the main study. Suitable patients treated with rituximab were added for this analysis. As there was a “post-hoc” analysis, we included patients with available data.
Following the reviewer’s advice, the figure with a study flowchart was added.
2. The Authors should specify how they considered an AE to be related to vaccination.
Authors’ reply:
Adverse events were reported when they occurred at the injection site or when they occurred in temporal proximity to vaccination. (This sentence was added in the methods section.) The reference list of adverse events was used as reported in the clinical trials of the mRNA BNT162b2 pivotal trials conducted in the general population (ref: Polack FP N Engl J Med 2020; 383:2603-2615).
Adverse events rate was comparable between the study groups as stated in the results section:
Vaccine responders and non-responders had a similar profile and rate of vaccine-related adverse events (supplementary table S3).
3. Given age or gender as an independent factor associated with immunogenicity to COVID-19 vaccination, it would be better if the authors are reporting biological sex or gender in your research.
Authors’ reply:
We have accordingly replaced the gender terminology with sex through the manuscript.
4. It would be interesting that the authors could perform survival analysis such as Kaplan-Meier analysis and Cox regression analysis based on an optimal cut-off value of predictor as the proposed prediction algorithm in this research.
Authors’ reply:
We thank the reviewer for the suggestion. As the study includes only a short-term follow up of vaccination response, the use of Kaplan-Meier analysis and Cox regression analysis would be of a particular importance in a setting of a longitudinal follow-up study.
5. As described in “Methods” section, an independent prospective cohort (n=48) from the Rambam Medical Health Care Campus was used for the validation of the proposed prediction model. The authors may present the related results and have a relevant discussion.
Authors’ reply:
The characteristics of the validation and main study groups were similar for age, sex, and the rate of seropositive response to vaccination. The characteristics of the cohort are presented in the supplementary table S2 and further elaborated in the discussion:
“The moderate diagnostic power of the model can be explained, at least partially, by the small size of the validation cohort that was characterized by a different representation of AIIRD compared to the main cohort. The fact that the immunogenic response was evaluated by different laboratory assays in both cohorts should not significantly affect the results since both assays correlated well with anti-SARS-CoV-2 neutralizing antibody levels.”
6. It would be clearly that the authors could add an additional figure to illustrate the comparisons of anti-S/RBD-IgG positive rates and titers among different groups.
Authors’ reply:
We have now added a corresponding Figure 2. Immunogenic response to the two-dose regimen of BNT162b2 vaccine in patients treated with rituximab according to AIIRD diagnosis.
Panel A: Anti-S1/S2 IgG titers (mean ± standard deviation)
Panel B: Rate of a seropositive response.
7. Some typing errors were noted. For example, in page 5, line 221-222, “Predictors associated with seropositive immunogenic response to the BNT162b2 vaccine in AIRD patients treated with RTX”.
Author’ reply:
Thank you for this comment. The cited sentence is a heading of a section and has now been clearly delineated.
Reviewer 2 Report
The authors aimed to identify predictors of a positive immunogenic response to the BNT162b2 mRNA vaccine in patients with autoimmune inflammatory rheumatic diseases (AIIRD) treated with RTX (AIIRD-RTX). For this purpose, were analyzed 108 AIIRD-RTX patients and 122 immunocompetent controls vaccinated with BNT162b2 mRNA participating in a multicenter vaccination study. The immunogenicity was defined by positive anti-SARS-CoV-2 S1/S2 IgG.
The study covers some issues that have been overlooked in other similar topics. The structure of the manuscript appears adequate and well divided in the sections. Moreover, the study is easy to follow, but some issues should be improved. Some of the comments that would improve the overall quality of the study are:
- Authors must pay attention to the technical terms acronyms they used in the text.
- English language needs to be revised.
- Conclusion Section: This paragraph needs to be improved. Please also add some "take-home message".
Author Response
The authors aimed to identify predictors of a positive immunogenic response to the BNT162b2 mRNA vaccine in patients with autoimmune inflammatory rheumatic diseases (AIIRD) treated with RTX (AIIRD-RTX). For this purpose, were analyzed 108 AIIRD-RTX patients and 122 immunocompetent controls vaccinated with BNT162b2 mRNA participating in a multicenter vaccination study. The immunogenicity was defined by positive anti-SARS-CoV-2 S1/S2 IgG.
The study covers some issues that have been overlooked in other similar topics. The structure of the manuscript appears adequate and well divided in the sections. Moreover, the study is easy to follow, but some issues should be improved. Some of the comments that would improve the overall quality of the study are:
Authors must pay attention to the technical terms acronyms they used in the text.
English language needs to be revised.
Authors’ reply:
The entire paper was checked by the institutional medical copyeditor.
Conclusion Section: This paragraph needs to be improved. Please also add some "take-home message".
Authors’ reply:
We have edited the conclusion section to underline the main take-home messages of the study:
This large multicenter study based on a diverse AIIRD population treated with RTX identified clinical predictors for a seropositive immunogenic vaccination response to BNT162b2 vaccination: a diagnosis of RA (as opposed to AAV and IIM), a low number of total RTX courses prior to vaccination, a high serum total IgG level prior to the last RTX course, and an extended interval between the RTX treatment and vaccination. Based on these variables, a prediction calculator for assessing the probability of seropositive response to the BNT162b2 vaccination was developed and validated, with a high sensitivity and good negative predictive value. The calculator can be useful for predicting the failure to respond to vaccination, thus suggesting to postpone vaccination. Taken together, the use of the predicting calculator might guide clinicians for optimal timing of BNT162b2 mRNA vaccination in AIIRD-RTX patients. Further investigation of predictors of vac-cine-induced response and vaccine efficacy as well as validation of the proposed prediction algorithm in prospective cohorts are warranted.
Reviewer 3 Report
The authors analyzed a group of RTX treated AIIRD patients vaccinated with the two-dose BNT162b2 mRNA vaccine regimen to identify predicting factors for a positive immunogenic response to vaccination and develop a predicting calculator. The stepwise backward multivariate logistic regression analysis indicated that ANCA-associated vasculitis, inflammatory myositis, higher IgG level prior to last RTX course, and extended interval between RTX treatment and vaccination were independent predictors associated with seropositive immunogenic response. In addition, the authors constructed a calculator predicting the probability of a seropositive immunogenic response, which performed with 90.5% sensitivity, 59.3% specificity, 63.3% positive and 88.9% negative predictive values. Although these results are interesting, there are several concerns as mentioned below for better understanding of this study.
- Based on the results of comparison between vaccine responders and non-responders, it seems reasonable to include age in the multivariate logistic regression analysis. The authors are recommended to perform such additional analysis or explain the reason why age was excluded.
- How was the rule for leaving the variable in the model determined? If the rule is p <0.05, is accuracy of the predicting calculator model improved or deteriorated?
- The authors are recommended to show how to make effective use of the proposed predicting calculator in clinical practice. This model with cut-off value being 0.41 has an excellent sensitivity and a good negative predictive value, meaning that the model is useful for predicting the failure of a seropositive immunogenic response and deciding to postpone vaccination. On the other hand, because the model has an inferior specificity and a relatively low positive predictive value, it seems difficult to accurately predict the probability of a seropositive immunogenic response.
Author Response
The authors analyzed a group of RTX treated AIIRD patients vaccinated with the two-dose BNT162b2 mRNA vaccine regimen to identify predicting factors for a positive immunogenic response to vaccination and develop a predicting calculator. The stepwise backward multivariate logistic regression analysis indicated that ANCA-associated vasculitis, inflammatory myositis, higher IgG level prior to last RTX course, and extended interval between RTX treatment and vaccination were independent predictors associated with seropositive immunogenic response. In addition, the authors constructed a calculator predicting the probability of a seropositive immunogenic response, which performed with 90.5% sensitivity, 59.3% specificity, 63.3% positive and 88.9% negative predictive values. Although these results are interesting, there are several concerns as mentioned below for better understanding of this study.
1. Based on the results of comparison between vaccine responders and non-responders, it seems reasonable to include age in the multivariate logistic regression analysis. The authors are recommended to perform such additional analysis or explain the reason why age was excluded.
Authors’ reply:
Age was marginally significant in the univariate model, so it was originally included at the backwards modeling process. In the subsequent multivariate, age was above p-value of 0.2, which was the exclusion criteria. Adding a non-significant covariate to an already packed model would have risked overfitting the model.
2. How was the rule for leaving the variable in the model determined? If the rule is p <0.05, is accuracy of the predicting calculator model improved or deteriorated?
Authors’ reply:
The rule for leaving the variable was determined as p<0.2 to allow a more flexible rule of the model to provide higher chances for the covariates to be eventually included. Therefore, the predicting model has been mostly related to the most significant covariates, while less significant covariates seem to have a marginal contribution on the predicting outcome.
3. The authors are recommended to show how to make effective use of the proposed predicting calculator in clinical practice. This model with cut-off value being 0.41 has an excellent sensitivity and a good negative predictive value, meaning that the model is useful for predicting the failure of a seropositive immunogenic response and deciding to postpone vaccination. On the other hand, because the model has an inferior specificity and a relatively low positive predictive value, it seems difficult to accurately predict the probability of a seropositive immunogenic response.
Authors’ reply:
We thank the reviewer for the insight. We have accordingly modified the discussion and conclusion sections.
Discussion:
Line _: As the model has a high sensitivity and a good negative predictive value, it is particular useful for predicting the failure to respond to vaccination, thus suggesting to postpone vaccination. In view of a relatively low positive predictive value, the prediction of the seropositive immunogenic response might be inaccurate.
Conclusions
This large multicenter study based on a diverse AIIRD population treated with RTX identified clinical predictors for a seropositive immunogenic vaccination response to BNT162b2 vaccination: a diagnosis of RA (as opposed to AAV and IIM), a low number of total RTX courses prior to vaccination, a high serum total IgG level prior to the last RTX course, and an extended interval between the RTX treatment and vaccination. Based on these variables, a prediction calculator for assessing the probability of seropositive response to the BNT162b2 vaccination was developed and validated, with a high sensitivity and good negative predictive value. The calculator can be useful for predicting the failure to respond to vaccination, thus suggesting to postpone vaccination. Taken together, the use of the predicting calculator might guide clinicians for optimal timing of BNT162b2 mRNA vaccination in AIIRD-RTX patients.